# Solutions of a three-dimensional multi-term fractional anomalous solute transport model for contamination in groundwater

**Imtiaz Ahmad**[1]*, **Ihteram Ali**[2], **Rashid Jan**[3], **Sahar Ahmed Idris**[4], **Mohamed Mousa**[5]*

**1** Institute of Informatics and Computing in Energy (IICE), Universiti Tenaga Nasional, Kajang, Selangor, Malaysia, **2** Department of Mathematics, Women University Swabi, Swabi, Pakistan, **3** Department of Civil Engineering, College of Engineering, Institute of Energy Infrastructure (IEI), Universiti Tenaga Nasional (UNITEN), Putrajaya Campus, Kajang, Selangor, Malaysia, **4** Faculty of Engineering, Department of Industrial Engineering, King Khalid University, Abha, Saudi Arabia, **5** Electrical Engineering Department, Future University in Egypt, Cairo, Egypt

* imtiazkakakhil@gmail.com (IA); mohamed.mossa@fue.edu.eg (MM)

**Data Availability Statement:** The authors confirm that the data supporting the findings of this study are available within the article.

## Abstract

The study presents a meshless computational approach for simulating the three-dimensional multi-term time-fractional mobile-immobile diffusion equation in the Caputo sense. The methodology combines a stable Crank-Nicolson time-integration scheme with the definition of the Caputo derivative to discretize the problem in the temporal direction. The spatial function derivative is approximated using the inverse multiquadric radial basis function. The solution is approximated on a set of scattered or uniform nodes, resulting in a sparse and well-conditioned coefficient matrix. The study highlights the advantages of meshless method, particularly their simplicity of implementation in higher dimensions. To validate the accuracy and efficacy of the proposed method, we performed numerical simulations and compared them with analytical solutions for various test problems. These simulations were carried out on computational domains of both rectangular and non-rectangular shapes. The research highlights the potential of meshless techniques in solving complex diffusion problems and its successful applications in groundwater contamination and other relevant fields.

## Introduction

Fractional calculus is a mathematical discipline exploring fractional derivatives and integrals and their properties and applications. By encompassing non-integer orders, it extends conventional calculus, enabling the examination of intricate physical phenomena beyond the scope of integer-order models. This field finds practical utility in engineering and various applied sciences, including finance, electromagnetism, signal processing, control theory and mechanics [1–8]. The concept of fractional calculus allows the order of a partial differential equation to vary with respect to time or space. As a result, the application of fractional calculus in differential equations has been broadened significantly. Fractional differential equations (FDEs) represent a broader category of conventional differential equations, accommodating derivatives of real or complex orders. These equations find extensive application in various areas, such as

**Funding:** The authors extend their appreciation to Deanship of Scientific Research at king Khalid university for funding this work through Large Groups (Project under grant number 2/478/44).

**Competing interests:** The authors have declared that no competing interests exist.

fluid mechanics, heat transfer, and electromagnetism [9–11]. Fractional partial differential equations (FPDEs), specifically, offer distinct advantages by providing a more accurate depiction of real-world phenomena that conventional DEs cannot adequately capture. Notably, FPDEs have been employed to model complex and nonlinear behavior in non-Newtonian fluids [12, 13]. Multi-term fractional order systems exhibit numerous advantages over single-term fractional order systems. Firstly, they enable more accurate modeling of intricate physical processes and phenomena due to their capability to encompass a broader spectrum of behaviors and dynamics [14]. Secondly, multi-term fractional order systems are particularly well-suited for control applications, as they provide improved stability margins and incredible robustness. Consequently, they effectively help disturbances and uncertainties in the system, resulting in improved performance and heightened reliability [15]. Moreover, multi-term fractional order systems offer enhanced efficiency and effectiveness in addressing engineering challenges, rendering them a valuable instrument across various applications, such as biomedical engineering, control systems, and signal processing [15–17]. In this study, we numerically simulate a three-dimensional (3D) multi-term time-fractional mobile-immobile diffusion equation (FMIDE), which is given as [18].

$$\frac{\partial \mathcal{V}(\bar{\mathbf{r}}, \tau)}{\partial \tau} + \mu \sum_{k=1}^{d} \frac{\partial^{\gamma_k} \mathcal{V}(\bar{\mathbf{r}}, \tau)}{\partial \tau^{\gamma_k}} = \alpha \nabla \mathcal{V}(\bar{\mathbf{r}}, \tau) + F(\bar{\mathbf{r}}, \tau) \equiv \mathcal{L} \mathcal{V}(\bar{\mathbf{r}}, \tau), \quad \tau > 0, \ \bar{\mathbf{r}} \in \Omega \subset \mathbb{R}^n \tag{1}$$

$$0 < \gamma_k \leq 1, k = 1, 2, 3, \ldots, d, \quad d \in \mathbb{N},$$

with conditions

$$\mathcal{V}(\bar{\mathbf{r}}, 0) = 0, \tag{2}$$

$$\mathcal{V}(\bar{\mathbf{r}}, \tau) = g_1(\bar{\mathbf{r}}, \tau), \quad \bar{\mathbf{r}} \in \partial \Omega, \tag{3}$$

In the context of the given problem, $\nabla$ denotes the Laplacian operator, and $\mu$ and $\alpha$ are known constants. Additionally, $F(\bar{\mathbf{r}}, \tau)$ and $g_1(\bar{\mathbf{r}}, \tau)$ are specified functions. Furthermore, the Caputo fractional derivative operator $\frac{\partial^{\gamma_k}}{\partial \tau^{\gamma_k}}$ applies to the function $\mathcal{V}(\bar{\mathbf{r}}, \tau)$, defined by Caputo [19] as follows

$$\frac{\partial^{\gamma_k} \mathcal{V}(\bar{\mathbf{r}}, \tau)}{\partial \tau^{\gamma_k}} = \begin{cases} \frac{1}{\Gamma(1 - \gamma_k)} \int_0^\tau (\tau - \zeta)^{-\gamma_k} \mathcal{V}_\zeta(\bar{\mathbf{r}}, \zeta) d\zeta, & 0 < \gamma_k < 1 \\ \\ \frac{\partial \mathcal{V}(\bar{\mathbf{r}}, \tau)}{\partial \tau}, & \gamma_k = 1. \end{cases}$$

whereas $\Gamma(.)$ represent the gamma function. The stability and convergence analysis of the fractional immobile/mobile transport model (1)–(3) has been investigated in previous works [18, 20].

The fractional diffusion equation characterizes anomalous diffusion phenomena, wherein particles do not disperse uniformly over time but instead reveal long-range correlations, non-uniform spreading, and other intricate behavior. It elucidates such phenomena through the extension of the classical diffusion equation to incorporate fractional spatial and temporal derivatives. The mobile-immobile fractional diffusion equation, a particular instance of sub-diffusion phenomena, has been extensively employed across various domains for simulating particle transport within heterogeneous media, where particles exhibit varying rates of movement between immobile and mobile regions [21]. This equation shows distinct advantages in

the characterization of anomalous solute transport phenomena within groundwater and porous media. Specifically, it effectively captures non-uniform transport behavior resulting from the presence of reactive surfaces and discrete flow pathways [22]. The movement of dissolved substances in liquids through porous media like groundwater or soil, is mathematically described by a model that accounts for the complexity influenced by diffusion, advection, and chemical reactions. This process is not straightforward or predictable [23, 24].

The anomalous solute transport model has proven to be highly versatile and applicable in various domains. These applications include predicting the movement of contaminants in groundwater and devising effective remediation strategies [25]. It also aids in understanding water movement in aquifers and surface water systems, optimizing pumping and recharge rates, and preventing aquifer depletion. Furthermore, the model facilitates the prediction of solvents and contaminant movement in soil, which is crucial for identifying effective remediation approaches. Additionally, it contributes to understanding carbon dioxide migration in subsurface formations, assisting in the development of efficient carbon sequestration strategies to mitigate climate change [25]. Researchers have explored the utilization of the mobile and immobile technique to address vehicular challenges in both saturated and unsaturated environments [23]. Gao et al. [24] presented experimental findings that showcased the efficacy of segregating the liquid phase within a porous medium into two distinct regions: mobile and immobile. This novel approach facilitates the representation of scale-dependent dispersion phenomena in heterogeneous porous media, thereby enabling a more comprehensive depiction of reactive solute transport processes.

Currently, there has been a notable increase in the investigation of FPDEs which are notoriously difficult to solve exactly. Consequently, a substantial portion of research efforts has been directed towards the development of numerical techniques for their solution [26–30]. Several widely employed approaches encompass employing finite difference or finite element methodologies, alongside fractional calculus techniques such as the Riemann-Liouville or Caputo operators. Researchers are currently focused on developing more efficient numerical methods such as variational iteration methods [31], spectral methods [32] and radial basis function (RBF) based meshless methods (MM) [33].

The utilization of meshless techniques has garnered significant attention in the scientific community due to their special capability to handle intricate geometries and yield accurate solutions without necessitating mesh generation. These numerical approaches do not need fixed mesh; instead, they utilize uniform or scattered data points to discretize the computational domain. Notably, RBF based MM have emerged as a prominent sub-type. These methods employ mathematical functions to effectively interpolate data values at scattered points, providing reliable approximations for solving PDEs [34–39]. RBF-based meshless methods offer a significant benefit in terms of their ease of implementation and adaptability to handle problems involving irregular geometries. Moreover, RBFs exhibit a 'global' characteristic, wherein the shape functions utilized for approximating solutions span the entire computational domain. Despite the usefulness of RBF-based meshless methods in solving diverse PDEs in engineering and science fields, some challenges exist. One such challenge pertains to the selection of suitable values for RBF parameters, specifically shape and size, which significantly impact the accuracy and efficiency of the methods. Moreover, another important limitation is the substantial computational cost associated with tackling large-scale problems, as these approaches involve solving dense systems of linear equations. Nevertheless, RBF based MM persist as a popular tool for PDE solving [40–42].

Local RBF methods [43] have been introduced as a viable approach to address the challenges encountered by global RBF methods. The key advantage of local RBFs lies in their ability to exhibit compact support centered around each individual data point. Consequently, the

resulting sparse matrices are better conditioned, facilitating the selection of an appropriate shape parameter and leading to a more accurate and efficient solution of the linear equations. As a result, local RBF methods tend to exhibit better accuracy and computational efficiency in large-scale problems.

## Motivation

The difficulty in solving fractional partial differential equations analytically motivates researchers to explore efficient numerical alternatives. PDEs play a crucial role in real-world applications such as engineering, physics, and finance, but finding exact analytical solutions is often impractical due to their nonlinearity and complexity. Consequently, various numerical methods have been developed and evaluated, tailored to specific problem characteristics and computational requirements. Traditional finite difference and finite element methods are widely used, but they may face challenges when dealing with irregular geometries, complex domains, and moving boundaries. In contrast, meshless methods present an attractive alternative as they do not require a pre-defined mesh and can efficiently handle complex geometries and unstructured domains. This adaptability makes meshless methods particularly suitable for fluid dynamics, structural mechanics, and data-driven modeling. This article introduces a meshless numerical scheme for solving PDE models using RBFs to compute spatial derivatives, ensuring accurate representation of unknown functions in higher dimensions. Additionally, the temporal direction is discretized using the Caputo derivative definition. The proposed meshless approach offers several advantages over traditional methods, including eliminating the need for a structured grid, ease of implementation for complex domains, and enabling seamless extension to multidimensional problems. Furthermore, the scheme demonstrates high accuracy and numerical stability, which are crucial for reliable and robust simulations.

## Fractional calculus: Analyzing the theoretical foundations of a time discrete scheme

In the context of discretizing the time variable, we begin by introducing key preliminary concepts from functional analysis. Fractional derivatives, an essential aspect of fractional calculus, hold a significant role in this process. To better understand their significance, we outline the fundamental fractional operator definitions that are commonly employed.

### Definition 1

The Riemann-Liouville fractional operator [44, 45]

$$\frac{\partial^{\gamma_k} \mathcal{V}(\bar{\mathbf{r}}, \tau)}{\partial \tau^{\gamma_k}} = \frac{1}{\Gamma(1 - \gamma_k)} \frac{d}{d\tau} \int_{\tau}^{T} \frac{(\mathcal{V}(\bar{\mathbf{r}}, \vartheta) - \mathcal{V}(\bar{\mathbf{r}}, T))}{(\vartheta - \tau)^{\gamma_k}} d\vartheta, \quad 0 < \gamma_k < 1. \tag{4}$$

### Definition 2

The Caputo's fractional operator [19]

$$\frac{\partial^{\gamma_k} \mathcal{V}(\bar{\mathbf{r}}, \tau)}{\partial \tau^{\gamma_k}} = \frac{1}{\Gamma(1 - \gamma_k)} \int_{0}^{\tau} \frac{\partial \mathcal{V}(\bar{\mathbf{r}}, \zeta)}{\partial \zeta} (\tau - \zeta)^{-\gamma_k} d\zeta, \quad 0 < \gamma_k < 1. \tag{5}$$

### Definition 3

The Atangana and Baleanu fractional operator [46]

$$
{}_{a}^{ABC}\frac{\partial^{\gamma_k}\mathcal{V}(\bar{\mathbf{r}},\tau)}{\partial\tau^{\gamma_k}} = \frac{B(\gamma_k)}{1-\gamma_k}\int_{a}^{\tau}\mathcal{V}'(\bar{\mathbf{r}})E_{\gamma_k}\left(-\frac{\gamma_k(\tau-\bar{\mathbf{r}})^{\gamma_k}}{1-\gamma_k}\right)d\bar{\mathbf{r}}, \quad 0 < \gamma_k < 1.
$$
(6)

**Definition 4**

He's fractional operator [47]

$$
\frac{\partial^{\gamma_k}\mathcal{V}(\bar{\mathbf{r}},\tau)}{\partial\tau^{\gamma_k}} = \frac{1}{\Gamma(1-\gamma_k)}\frac{d}{d\bar{\mathbf{r}}}\int_{\tau_0}^{\tau}(\tau-\zeta)^{-\gamma_k}[\mathcal{V}_0(\zeta)-\mathcal{V}(\zeta)], \quad 0 < \gamma_k < 1.
$$
(7)

## Introduction to applied functional analysis: A preliminary overview

Consider a bounded and open domain $\Omega$ in $\mathbb{R}^2$, where $d\bar{\mathbf{r}}$ denotes the Lebesgue measure on $\mathbb{R}^2$. For $p < \infty$, we define the space $L^p(\Omega)$ as the set of measurable functions $\mathcal{V} : \Omega \to \mathbb{R}$ satisfying $\int_{\Omega}|\mathcal{V}(\bar{\mathbf{r}})|^p d\bar{\mathbf{r}} \leq \infty$. More generally, we can represent this Banach space by its norm.

$$
\| \mathcal{V} \|_{L^p(\Omega)} = \left(\int_{\Omega}|\mathcal{V}(\bar{\mathbf{r}})|^p d\bar{\mathbf{r}}\right)^{\frac{1}{p}}.
$$

The Hilbert space $L^p(\Omega)$ possesses an inner product given by

$$
(\mathcal{V},\mathcal{W}) = \int_{\Omega}\mathcal{V}(\bar{\mathbf{r}})\mathcal{W}(\bar{\mathbf{r}})d\bar{\mathbf{r}},
$$

using the prescribed norm in $L^2$

$$
\| \mathcal{V} \|_{2} = [(\mathcal{V},\mathcal{V})]^{\frac{1}{2}} = \left[\int_{\Omega}\mathcal{V}(\bar{\mathbf{r}})\mathcal{V}(\bar{\mathbf{r}})d\bar{\mathbf{r}}\right]^{\frac{1}{2}}.
$$

Also we suppose that $\Omega$ is an open domain in $\mathbb{R}^d$, $\alpha = (\alpha_1, \ldots, \alpha_d)$ is a $d$-tuple of non-negative integers and $|\alpha| = \sum_{i=1}^{p}\alpha_i$. Accordingly, we put

$$
D^{\alpha}\mathcal{W} = \frac{\partial^{|\alpha|}\mathcal{W}}{\partial\bar{\mathbf{r}}_1^{\alpha}\partial\bar{\mathbf{r}}_2^{\alpha}\ldots\partial\bar{\mathbf{r}}_d^{\alpha}}.
$$

In this regard, one can obtain

$$
\begin{aligned}
H^1(\Omega) &= \left\{\mathcal{W} \in L^2(\Omega), \quad \frac{d\mathcal{W}}{d\bar{\mathbf{r}}} \in L^2(\Omega)\right\}, \\
H_0^1(\Omega) &= \left\{\mathcal{W} \in H^1(\Omega), \quad \mathcal{W}|_{\partial(\Omega)} = 0\right\}, \\
H^m(\Omega) &= \mathcal{W} \in L^2(\Omega), \quad D^{\alpha}\mathcal{W} \in L^2(\Omega) \quad \text{for all positive integer } |\alpha| \leq m.
\end{aligned}
$$
(8)

Next, we introduce the elucidation of inner product within a Hilbert space.

$$
(\mathcal{V},\mathcal{W})_m = \sum_{|\alpha|\leq m}\int_{\Omega}D^{\alpha}\mathcal{V}(\bar{\mathbf{r}})D^{\alpha}\mathcal{W}(\bar{\mathbf{r}})d\bar{\mathbf{r}},
$$

Inducing the norm

$$\| \mathcal{V} \|_{H^m(\Omega)} = \left( \sum_{|\alpha| \le m} \| D^\alpha \mathcal{V} \|_{L^2(\Omega)}^2 \right)^{\frac{1}{2}}.$$

The Sobolev space $X^{1,p}(I)$ is said to be

$$X^{1,p}(I) = \left\{ \mathcal{V} \in L^p(I); \quad \exists g \in L^p(I) : \int_I \mathcal{V} \varphi' = \int_I g \varphi', \quad \forall \varphi \in C^1(I) \right\}.$$

Additionally, within the scope of this manuscript, we introduce the ensuing inner product alongside the corresponding energy norms within the function spaces $L^2$ and $H^1$.

$$\| \mathcal{W} \| = (\mathcal{W}, \mathcal{W})^{1/2}, \quad \| \mathcal{W} \|_1 = (\mathcal{W}, \mathcal{W})_1^{1/2}, \quad |\mathcal{W}|_1 = \left( \frac{\partial \mathcal{W}}{\partial \bar{\mathbf{r}}}, \frac{\partial \mathcal{W}}{\partial \bar{\mathbf{r}}} \right)^{1/2},$$

by inner products of $L^2(\Omega)$ and $H^1(\Omega)$

$$(\mathcal{V}, \mathcal{W}) = \int U(\bar{\mathbf{r}}) \mathcal{W}(\bar{\mathbf{r}}) d\bar{\mathbf{r}}, \quad (\mathcal{V}, \mathcal{W})_1 = (\mathcal{V}, \mathcal{W}) + \left( \frac{\partial \mathcal{V}}{\partial \bar{\mathbf{r}}}, \frac{\partial \mathcal{W}}{\partial \bar{\mathbf{r}}} \right),$$

respectively.

Let us define $\mathfrak{J} = \frac{T}{M}$ be the mesh size in time, and $\tau_n = n\mathfrak{J}, n \in \mathbb{N}^+$, are the total $M$ temporal discretization points.

**Lemma 1.** Let us suppose $\eta(t) \in C^2[0, T]$ and $0 < \gamma_k < 1$, then it holds that

$$\int_0^{\tau_n} \eta'(\bar{\mathbf{r}})(\tau_n - \bar{\mathbf{r}})^{-\gamma_k} d\bar{\mathbf{r}} = \sum_{j=1}^n \frac{\eta(\tau_j) - \eta(\tau_{j-1})}{\mathfrak{J}},$$

$$\int_{\tau_{j-1}}^{\tau_j} (\tau_n - \bar{\mathbf{r}})^{-\gamma_k} d\bar{\mathbf{r}} + R^n, \quad 1 \le n \le M,$$

and $|R^n| \le \left( \frac{1}{2(1-\gamma_k)} + \frac{1}{2} \right) \mathfrak{J}^{2-\gamma_k} \max_{0 \le \tau \le \tau_n} |\eta''(t)|$.

**Proof.** Sun et al. [48].

**Lemma 2.** Let $0 < \gamma_k < 1$, $a_0 = \frac{1}{\mathfrak{J}\Gamma(1-\gamma_k)}$ and $b_j = \frac{\mathfrak{J}^{1-\gamma_k}}{(1-\gamma_k)} \left[ (j+1)^{1-\gamma_k} - (j)^{1-\gamma_k} \right]$, then

$$\frac{1}{\Gamma(1-\gamma_k)} \int_0^{\tau_n} \frac{\eta'(\bar{\mathbf{r}})}{(\tau_n - \bar{\mathbf{r}})^{\gamma_k}} d\bar{\mathbf{r}} - a_0 \left[ b_0 \eta(\tau_n) - \sum_{j=1}^{n-1} \left( b_{n-j-1} - b_{n-j} \right) \eta(\tau_j) - b_{n-1} \eta(0) \right]$$

$$\le \frac{1}{2\Gamma(1-\gamma_k)} \left( 1 + \frac{1}{(1-\gamma_k)} \right) \mathfrak{J}^{2-\gamma_k} \max_{0 \le \tau \le \tau_i} |\eta''(\tau)|.$$

**Proof.** Directly follows from Lemma 1.

**Lemma 3.** Let $b_j = \frac{\mathfrak{J}^{1-\gamma_k}}{(1-\gamma_k)} \left[ (j+1)^{1-\gamma_k} - (j)^{1-\gamma_k} \right]$, where $0 < \gamma_k < 1, j = 0, 1, 2, \ldots$, then $b_0 > b_1 > b_2 > \ldots > b_j \to 0$, as $j \to \infty$.

**Proof.** Sun et al. [48].

## Formulation of space discretization

To approximate the derivatives of $\mathcal{V}(\bar{\mathbf{r}}, \tau)$ at the centers $\bar{\mathbf{r}}_i$, $\{\bar{\mathbf{r}}_{i_1}, \bar{\mathbf{r}}_{i_2}, \bar{\mathbf{r}}_{i_3}, \ldots, \bar{\mathbf{r}}_{i_{n_i}}\} \subset \{\bar{\mathbf{r}}_1, \bar{\mathbf{r}}_2, \ldots, \bar{\mathbf{r}}_{N^n}\}$, $n_i \ll N^n$, $i = 1, 2, \ldots, N^n$. In case of one-, two and three-dimensional $\bar{\mathbf{r}} = r$, $\bar{\mathbf{r}} = (r, s)$ and $\bar{\mathbf{r}} = (r, s, z)$ respectively.

The following elucidates the methodology concerning the one-dimensional scenario.

$$\mathcal{V}^{(m)}(r_h) \approx \sum_{k=1}^{n_h} \lambda_k^{(m)} \mathcal{V}(r_{hk}), \ h = 1, 2, \ldots, N. \tag{9}$$

Plugging RBF $\psi(\|r - r_p\|)$ in (9), we obtain

$$\psi^{(m)}(\| r_h - r_p \|) = \sum_{k=1}^{n_h} \lambda_{hk}^{(m)} \psi(\| r_{hk} - r_p \|), \ p = h1, h2, \ldots, hn_h, \tag{10}$$

In this case, the following matrix expression of Eq (10) is obtained by employed the inverse multiquadric (IMQ) RBF $\psi(\| r_{hk} - r_p \|) = 1/\sqrt{1 + (c \| r_{hk} - r_p\|)^2}$.

$$\underbrace{\begin{bmatrix} \psi_{h1}^{(m)}(r_h) \\ \psi_{h2}^{(m)}(r_h) \\ \vdots \\ \psi_{hn_h}^{(m)}(r_h) \end{bmatrix}}_{\psi_{n_h}^{(m)}} = \underbrace{\begin{bmatrix} \psi_{h1}(r_{h1}) & \psi_{h2}(r_{h1}) & \ldots & \psi_{hn_h}(r_{h1}) \\ \psi_{h1}(r_{h2}) & \psi_{h2}(r_{h2}) & \ldots & \psi_{hn_h}(r_{h2}) \\ \vdots & \vdots & \ldots & \vdots \\ \psi_{h1}(rh_{n_h}) & \psi_{h2}(r_{hn_h}) & \ldots & \psi_{hn_h}(rh_{n_h}) \end{bmatrix}}_{\mathbf{A}_{n_h}} \underbrace{\begin{bmatrix} \lambda_{h1}^{(m)} \\ \lambda_{h2}^{(m)} \\ \vdots \\ \lambda_{hn_h}^{(m)} \end{bmatrix}}_{\lambda_{n_h}^{(m)}}, \tag{11}$$

where

$$\psi_p(r_k) = \psi(\| r_k - r_p \|), \ p = h1, h2, \ldots, hn_h, \tag{12}$$

for each $k = i1, h2, \ldots, hn_h$. Eq (11) can be written as

$$\psi_{n_h}^{(m)} = \mathbf{A}_{n_h} \lambda_{n_h}^{(m)}, \tag{13}$$

As reported in [49], it has been established that the matrix $\mathbf{A}_{n_h}$ is invertible. Utilizing (13), we can derive the following result

$$\lambda_{n_h}^{(m)} = \mathbf{A}_{n_h}^{-1} \psi_{n_h}^{(m)}. \tag{14}$$

Eqs (9) and (14) implies

$$\mathcal{V}^{(m)}(r_h) = (\lambda_{n_h}^{(m)})^T \mathcal{V}_{n_h}, \tag{15}$$

where

$$\mathcal{V}_{n_h} = [\mathcal{V}(r_{h1}), \mathcal{V}(r_{h2}), \ldots, \mathcal{V}(r_{hn_h})]^T. \tag{16}$$

The following elucidates the methodology concerning the two-dimensional scenario.

$$\mathcal{V}_r^{(m)}(r_h, s_h) \approx \sum_{k=1}^{n_h} \rho_k^{(m)} \mathcal{V}(r_{hk}, s_{hk}), \ h = 1, 2, \ldots, N^2, \tag{17}$$

$$\mathcal{V}_y^{(m)}(r_h, s_h) \approx \sum_{k=1}^{n_h} \eta_k^{(m)} \mathcal{V}(r_{hk}, s_{hk}), \ h = 1, 2, \ldots, N^2. \tag{18}$$

The coefficients $\rho_k^{(m)}$ and $\eta_k^{(m)}$ ($k = 1, 2, \ldots, n_h$) can be determined through the following process

$$\rho_{n_h}^{(m)} = \mathbf{A}_{n_h}^{-1} \Theta_{n_h}^{(m)}, \tag{19}$$

$$\eta_{n_h}^{(m)} = \mathbf{A}_{n_h}^{-1} \Theta_{n_h}^{(m)}. \tag{20}$$

Similarly, the identical procedure can be extended to the three-dimensional scenario.

An approximate semi-discretized formulation for the mathematical model described by (1) with associated initial and boundary conditions is developed herein by employing the suggested meshless methodology.

$$\frac{\partial \mathcal{V}(\bar{\mathbf{r}}, \tau)}{\partial \tau} + \mu \sum_{k=1}^{d} \frac{\partial^{\gamma_k} \mathcal{V}(\bar{\mathbf{r}}, \tau)}{\partial \tau^{\gamma_k}} = \mathcal{D}\mathfrak{V} + \mathbf{k}(\tau), \ \mathfrak{v}(0) = \mathbf{b}, \tag{21}$$

Let $\mathcal{D}$ denote the sparse coefficient matrix derived from the recommended MM approximation. The vectors $\mathbf{b}$ and $\mathbf{k}$ of size $N \times 1$ represent the boundary and initial conditions, respectively, of the given problem.

## Time-stepping schemes

The time-fractional Caputo derivative for $0 < \gamma_k \leq 1$ is $\frac{\partial^{\gamma_k} \mathcal{V}(\bar{\mathbf{r}}, \tau)}{\partial \tau^{\gamma_k}}$, which can be written as

$$\frac{\partial^{\gamma_k} \mathcal{V}(\bar{\mathbf{r}}, \tau)}{\partial \tau^{\gamma_k}} = \begin{cases} \dfrac{1}{\Gamma(1 - \gamma_k)} \displaystyle\int_0^\tau \dfrac{\partial \mathcal{V}(\bar{\mathbf{r}}, \zeta)}{\partial \zeta} (\tau - \zeta)^{-\gamma_k} d\zeta, & 0 < \gamma_k < 1 \\[4mm] \dfrac{\partial \mathcal{V}(\bar{\mathbf{r}}, \tau)}{\partial \tau}, & \gamma_k = 1. \end{cases} \tag{22}$$

Taking into account $M + 1$ equidistant time levels $\tau_0, \tau_1, \ldots, \tau_M$ within the interval $[0, \tau]$, where the time step is denoted as $\mathfrak{J}$ and defined as $\tau_n = n\mathfrak{J}$ for $n = 0, 1, 2, \ldots, M$, we propose a first-order finite difference scheme to approximate the time fractional derivative term.

$$\begin{aligned} \frac{\partial^{\gamma_k} \mathcal{V}(\bar{\mathbf{r}}, \tau_{n+1})}{\partial \tau^{\gamma_k}} &= \frac{1}{\Gamma(1 - \gamma_k)} \int_0^{\tau_{n+1}} \frac{\partial \mathcal{V}(\bar{\mathbf{r}}, \zeta)}{\partial \zeta} (\tau_{n+1} - \zeta)^{-\gamma_k} d\zeta, \\[4mm] &= \frac{1}{\Gamma(1 - \gamma_k)} \sum_{j=0}^{n} \int_{j\mathfrak{J}}^{(j+1)\mathfrak{J}} \frac{\partial \mathcal{V}(\bar{\mathbf{r}}, \zeta_j)}{\partial \zeta} (\tau_{j+1} - \zeta)^{-\gamma_k} d\zeta, \end{aligned} \tag{23}$$

where $\frac{\partial \mathcal{V}(\bar{\mathbf{r}}, \zeta_j)}{\partial \zeta}$, is approximated as follow

$$\frac{\partial \mathcal{V}(\bar{\mathbf{r}}, \zeta_j)}{\partial \zeta} = \frac{\mathcal{V}(\bar{\mathbf{r}}, \zeta_{j+1}) - \mathcal{V}(\bar{\mathbf{r}}, \zeta_j)}{\zeta} + \mathcal{O}(\mathfrak{I}).$$

Then

$$\frac{\partial^{\gamma_k} \mathcal{V}(\bar{\mathbf{r}}, \tau_{n+1})}{\partial \tau^{\gamma_k}} = \frac{1}{\Gamma(1 - \gamma_k)} \sum_{j=0}^{n} \frac{\mathcal{V}(\bar{\mathbf{r}}, \tau_{j+1}) - \mathcal{V}(\bar{\mathbf{r}}, \tau_j)}{\mathfrak{I}} \int_{j\mathfrak{I}}^{(j+1)\mathfrak{I}} (\tau_{j+1} - \zeta)^{-\gamma_k} d\zeta,$$

$$= \frac{1}{\Gamma(1 - \gamma_k)} \sum_{j=0}^{n} \frac{\mathcal{V}(\bar{\mathbf{r}}, \tau_{n+1-j}) - \mathcal{V}(\bar{\mathbf{r}}, \tau_{n-j})}{\mathfrak{I}} \int_{j\mathfrak{I}}^{(j+1)\mathfrak{I}} (\tau_{j+1} - \zeta)^{-\gamma_k} d\zeta,$$

$$= \begin{cases} \dfrac{\mathfrak{I}^{-\gamma_k}}{\Gamma(2 - \gamma_k)} (\mathcal{V}^{n+1} - \mathcal{V}^n) + \dfrac{\mathfrak{I}^{-\gamma_k}}{\Gamma(2 - \gamma_k)} \sum_{j=1}^{n} (U\mathcal{V}^{n+1-j} - \mathcal{V}^{n-j})[(j+1)^{1-\gamma_k} - j^{1-\gamma_k}], & n \geq 1 \\[3mm] \dfrac{\mathfrak{I}^{-\gamma_k}}{\Gamma(2 - \gamma_k)} (\mathcal{V}^1 - \mathcal{V}^0), & n = 0. \end{cases}$$

Let $\alpha_0 = \frac{\mathfrak{I}^{-\gamma_k}}{\Gamma(2-\gamma_k)}$ and $b_j = (j+1)^{1-\gamma_k} - j^{1-\gamma_k}, j = 0, 1, \ldots, n$. The more precise form is

$$\frac{\partial^{\gamma_k} \mathcal{V}(\bar{\mathbf{r}}, \tau_{n+1})}{\partial t^{\gamma_k}} = \begin{cases} \alpha_0 (\mathcal{V}^{n+1} - \mathcal{V}^n) + \alpha_0 \sum_{j=1}^{n} b_j (\mathcal{V}^{n+1-j} - \mathcal{V}^{n-j}), & n \geq 1 \\[3mm] \alpha_0 (\mathcal{V}^1 - \mathcal{V}^0), & n = 0. \end{cases} \tag{24}$$

## Formulation of $\theta$-weighted scheme

The methodology for the time discretization of a 1-term time-fractional order utilizing the $\theta$-weighted scheme along (24) to approximate the model given by (1) (taking $\mu = 1$), we have

$$\frac{\partial \mathcal{V}}{\partial \tau} + \frac{\partial^{\gamma_k} \mathcal{V}}{\partial \tau^{\gamma_k}} \equiv \mathcal{L}\mathcal{V}, \tag{25}$$

now for $n \geq 1$

$$\frac{\mathcal{V}^{(n+1)} - \mathcal{V}^{(n)}}{\mathfrak{I}} + \alpha_0 \mathcal{V}^{(n+1)} - \alpha_0 \mathcal{V}^{(n)} + \alpha_0 \sum_{j=1}^{n} b_j (\mathcal{V}^{n+1-j} - \mathcal{V}^{n-j}) = \theta \mathcal{L}\mathcal{V}^{(n+1)} + (1 - \theta)\mathcal{L}\mathcal{V}^{(n)}, \tag{26}$$

we get

$$\mathcal{V}^{(n+1)} = (I + \mathfrak{I}\alpha_0 I - \mathfrak{I}\theta\mathcal{L})^{-1} \left( (I + \mathfrak{I}\alpha_0 I + \mathfrak{I}(1 - \theta)\mathcal{L})\mathcal{V}^{(n)} + \alpha_0 \sum_{j=1}^{n} b_j (\mathcal{V}^{n+1-j} - \mathcal{V}^{n-j}) \right), \tag{27}$$

similarly for $n = 0$

$$\mathcal{V}^{(1)} = (I + \mathfrak{I}\alpha_0 I - \mathfrak{I}\theta\mathcal{L})^{-1} ((I + \mathfrak{I}\alpha_0 I + \mathfrak{I}(1 - \theta)\mathcal{L})\mathcal{V}^{(0)}), \tag{28}$$

After implementing the proposed meshless technique (discussed in Section), (27)–(28) lead to

$$\mathcal{V}^{(n+1)} = (I - \Im\theta L)^{-1}\left((I + \Im(1-\theta)L)\mathcal{V}^{(n)} + \alpha_0\sum_{j=1}^{n}b_j(\mathcal{V}^{n+1-j} - \mathcal{V}^{n-j})\right), \tag{29}$$

$$\mathcal{V}^{(1)} = (I + \Im\alpha_0 I - \Im\theta L)^{-1}((I + \Im\alpha_0 I + \Im(1-\theta)L)\mathcal{V}^{(0)}), \tag{30}$$

Let $L$ represent the weight matrix of the differential operator $\mathcal{L}$, and $I$ denote an identity matrix. Eqs (29) and (30) reduces to Crank-Nicolson scheme for $\theta = \frac{1}{2}$.

Likewise, this process can be iterated for 2-, 3-, and 5-term time-fractional derivatives.

## Numerical simulation and discussion

This section presents a comprehensive analysis of the accuracy and applicability of the suggested meshless technique for approximating the numerical solution of the underlying problem given in (1). The proposed method utilizes IMQ with shape parameter value $c = 1$ and its effectiveness is assessed for different time fractional orders, including 2-term, 3-term, and 5-term equations. Furthermore, the proposed method is subjected to testing on both non-rectangular and rectangular domains. The temporal step size is set to $\Im = 0.0005$, and the spatial domain is defined as [0, 1], unless specified otherwise. The accuracy of the suggested method is evaluated utilizing the following criteria:

$$Absolute - error = |\hat{\mathbb{V}} - \mathcal{V}|,$$

$$MaxE = \max(Absolute - error),$$

$$RMS = \sqrt{\frac{\sum_{h=1}^{N^n}\left(\hat{\mathbb{V}}_i - \mathcal{V}_i\right)^2}{N}}, \tag{31}$$

where $\hat{\mathcal{V}}$ is the exact solution.

**Test Problem 1** *The closed-form solution for the model* (1), *with* $\mu = \alpha = 1$ *is*

$$\mathcal{V}(\bar{r}, \tau) = t^2\cos(2\pi r)\cos(2\pi s)\cos(2\pi z), \quad \bar{r} = (r, s, z) \in \Omega, \tag{32}$$

The presented numerical results for Problem 1 are obtained using the recommended meshless approach and are shown in Table 1. The approach uses various parameters, including the number of nodes $N$, temporal step size $\Im$, and fractional-orders i.e., $\gamma_1 = \gamma_2 = 0.5$ for 2-term, $\gamma_1 = \gamma_2 = \gamma_3 = 0.5$ for 3-term, and $\gamma_1 = \gamma_2 = \gamma_3 = \gamma_4 = \gamma_5 = 0.5$ for 5-term. Also, the final time $\tau$ is set at 0.5, while the accuracy is evaluated using the *MaxE* and *RMS*. The results indicate that the suggested meshless approach provides better accuracy. Moreover, the results in Table 1 reveal that as the number of nodes and the number of terms in the time-fractional orders increase, the accuracy improves. Numerical results for various combinations of $\tau$, $\Im$, and $\gamma_1 =$

**Table 1. The outcomes obtained from utilizing the meshless approach to address Problem 1.**

| | N = 5 | | N = 10 | | N = 15 | |
|---|---|---|---|---|---|---|
| | RMS | MaxE | RMS | MaxE | RMS | MaxE |
| 2-term | 5.1738e-03 | 5.1204e-02 | 1.2283e-03 | 7.2246e-03 | 5.9979e-04 | 3.5535e-03 |
| 3-term | 4.9708e-03 | 4.9624e-02 | 1.2002e-03 | 7.0372e-03 | 5.8792e-04 | 3.4721e-03 |
| 5-term | 4.6189e-03 | 4.6796e-02 | 1.1501e-03 | 6.6990e-03 | 5.6632e-04 | 3.3233e-03 |

**Table 2. The outcomes obtained from utilizing the meshless approach to address Problem 1.**

|  | $\mathfrak{J} = 0.01$ |  | $\mathfrak{J} = 0.001$ |  | $\mathfrak{J} = 0.0005$ |  |
|---|---|---|---|---|---|---|
|  | *RMS* | *MaxE* | *RMS* | *MaxE* | *RMS* | *MaxE* |
| 2-term | 3.7245e-03 | 2.9888e-02 | 7.1603e-04 | 3.7482e-03 | 5.9979e-04 | 3.5535e-03 |
| 3-term | 3.4222e-03 | 2.6513e-02 | 7.0084e-04 | 3.6707e-03 | 5.8792e-04 | 3.4721e-03 |
| 5-term | 2.9695e-03 | 2.1462e-02 | 6.7293e-04 | 3.5267e-03 | 5.6632e-04 | 3.3233e-03 |

$\gamma_2 = 0.5$, $\gamma_1 = \gamma_2 = \gamma_3 = 0.5$, and $\gamma_1 = \gamma_2 = \gamma_3 = \gamma_4 = \gamma_5 = 0.5$ with $N = 15$ nodes are presented in Tables 2 and 3. The results indicate that accuracy has been reasonably improved in this scenario. Figs 1 and 2 display a comparative analysis of 2-term, 3-term, and 5-term solutions. Fig 1 presents a comparative analysis between the numerical solutions and the corresponding exact solutions, along with their respective absolute errors. Additionally, Fig 2 exhibits the results obtained for the 5-term fractional order. The observations drawn from these figures strongly support that the recommended method yields accurate results.

**Test Problem 2** *The closed-form solution for the model* (1), *with* $\mu = \alpha = 1$ *is*

$$\mathcal{V}(\bar{r}, \tau) = \exp(-\tau) \sin(\pi r) \sin(\pi s) \sin(\pi z), \quad \bar{r} = (r, s, z) \in \Omega, \tag{33}$$

In Table 4, we display the results of numerical simulations concerning Problem 2 under different values of $N$, while fixed value of $\tau = 0.5$ and $\gamma_1 = \gamma_2 = 0.5$ for the 2-term case, $\gamma_1 = \gamma_2 = \gamma_3 = 0.5$ for the 3-term case, and $\gamma_1 = \gamma_2 = \gamma_3 = \gamma_4 = \gamma_5 = 0.5$ for the 5-term case. Similarly, in Table 5 numerical results for the number of $\mathfrak{J}$ and $\tau$, while keeping the nodes $N = 15$ and $\gamma_k = 0.5$ fixed. The value of $\gamma_k$ is chosen as $\gamma_k = \gamma_1 = \gamma_2$ for the 2-term case, and as $\gamma_k = \gamma_1 = \gamma_2 = \gamma_3$ and $\gamma_k = \gamma_1 = \gamma_2 = \gamma_3 = \gamma_4 = \gamma_5$ for the 3-term and 5-term cases, respectively. It should be noted that accurate numerical results have been obtained in all these cases. Figs 3 and 4 illustrate a comparative examination of solutions using 2-term, 3-term, and 5-term fractional orders. In Fig 3, we present a comparison between numerical and exact solutions, accompanied by their respective absolute errors. Whereas Fig 4 displays the results obtained through the 5-term fractional order approach. The observed results strongly suggest that the proposed method yields highly accurate results.

The meshless method stands out for its capability to tackle problems in non-uniform geometries. Unlike conventional techniques, it doesn't rely on a predetermined mesh structure, which is especially useful when dealing with intricate and irregular shapes. Moreover, this approach is versatile and adaptable, as it can effectively solve problems with scattered data points without needing any connectivity information. The effectiveness of this feature has been demonstrated on the computational domains [43] illustrated in Figs 5 and 6, with the corresponding results. These results indicate that the proposed meshless method produces precise outcomes, even in challenging scenarios.

**Table 3. The outcomes obtained from utilizing the meshless approach to address Problem 1.**

|  | $\tau = 0.1$ |  | $\tau = 1$ |  | $\tau = 1.5$ |  |
|---|---|---|---|---|---|---|
|  | *RMS* | *MaxE* | *RMS* | *MaxE* | *RMS* | *MaxE* |
| 2-term | 2.6929e-05 | 1.5593e-04 | 2.3850e-03 | 1.4205e-02 | 5.3653e-03 | 3.2024e-02 |
| 3-term | 2.6173e-05 | 1.5031e-04 | 2.3469e-03 | 1.3959e-02 | 5.2905e-03 | 3.1561e-02 |
| 5-term | 2.4821e-05 | 1.4038e-04 | 2.2763e-03 | 1.3500e-02 | 5.1504e-03 | 3.0686e-02 |

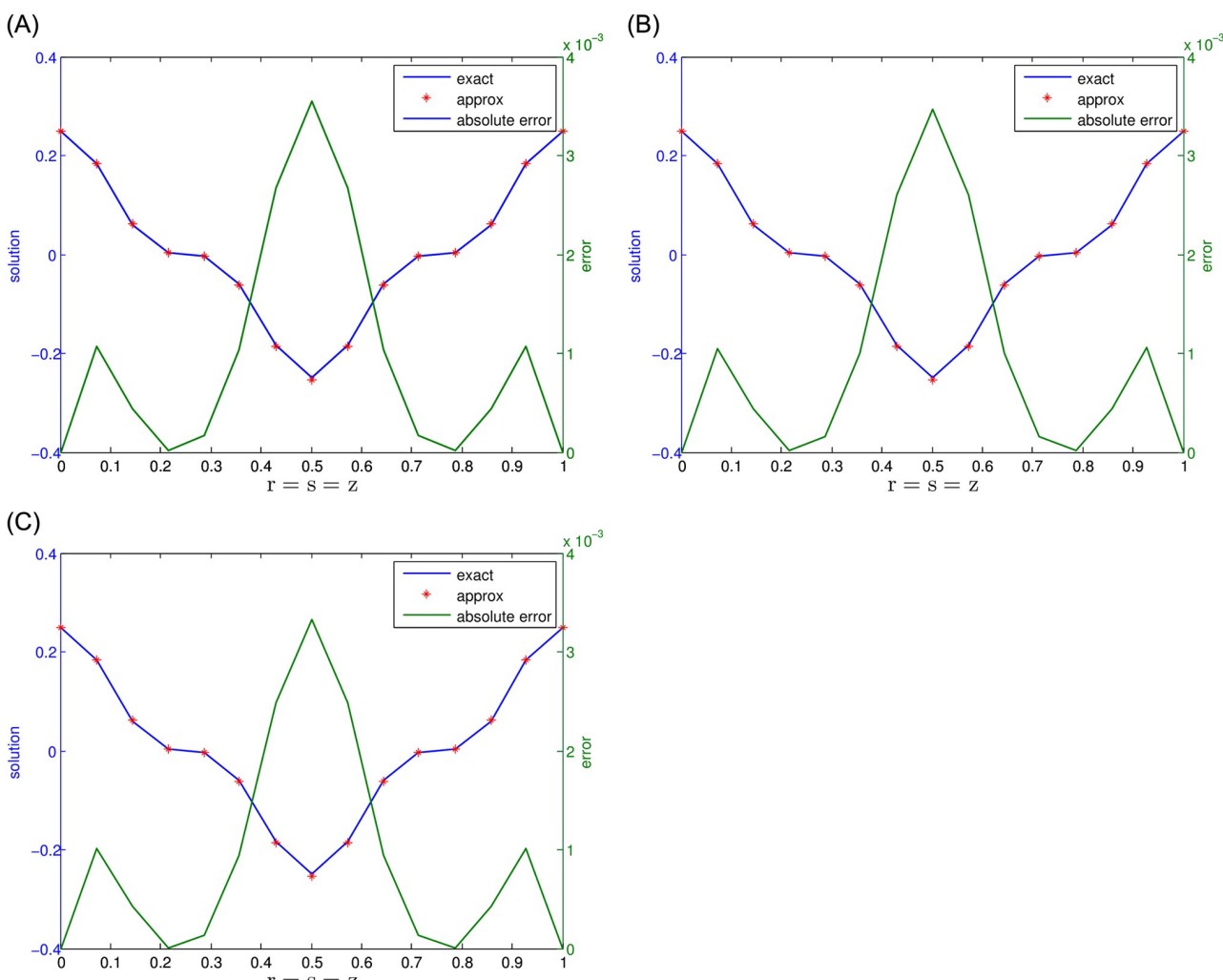

**Fig 1. For Problem 1, the results for 2-, 3-, and 5-term fractional derivatives are presented, indicating a comparison between exact and numerical outcomes along with the absolute error.**

**Test Problem 3** *The closed-form solution for the model* (1), *with* $\mu = \alpha = 1$ *is*

$$\mathcal{V}(\bar{\boldsymbol{r}}, \tau) = \tau^2 e^{(r+s+z)}, \quad \bar{\boldsymbol{r}} = (r, s, z) \in \Omega, \tag{34}$$

In Table 6 the numerical results regarding various fractional-order values $\gamma_k$, $N = 15$ and $\tau = 0.5$ are obtained. For the 2-term case, $\gamma_k = \gamma_1 = \gamma_2$, similarly for the 3- and 5-term cases $\gamma_k = \gamma_1 = \gamma_2 = \gamma_3$ and $\gamma_k = \gamma_1 = \gamma_2 = \gamma_3 = \gamma_4 = \gamma_5$, respectively. Also, Table 7 shows results computed for various values of $\mathfrak{J}$, $N$ and $\tau = 0.5$. In this case, the fractional-order $\gamma_1 = \gamma_2 = 0.5$ is used for the 2-term case, $\gamma_1 = \gamma_2 = \gamma_3 = 0.5$ for the 3-term case, and $\gamma_1 = \gamma_2 = \gamma_3 = \gamma_4 = \gamma_5 = 0.5$ for the 5-term case. The tables clearly demonstrate that increasing the number of terms in the time-fractional orders and the number of nodes results in improved accuracy. Fig 7 displays a comparative analysis of the numerical and exact solutions. The absolute error between these solutions is also given in the figure. The figure indicates that the proposed method provides accurate results.

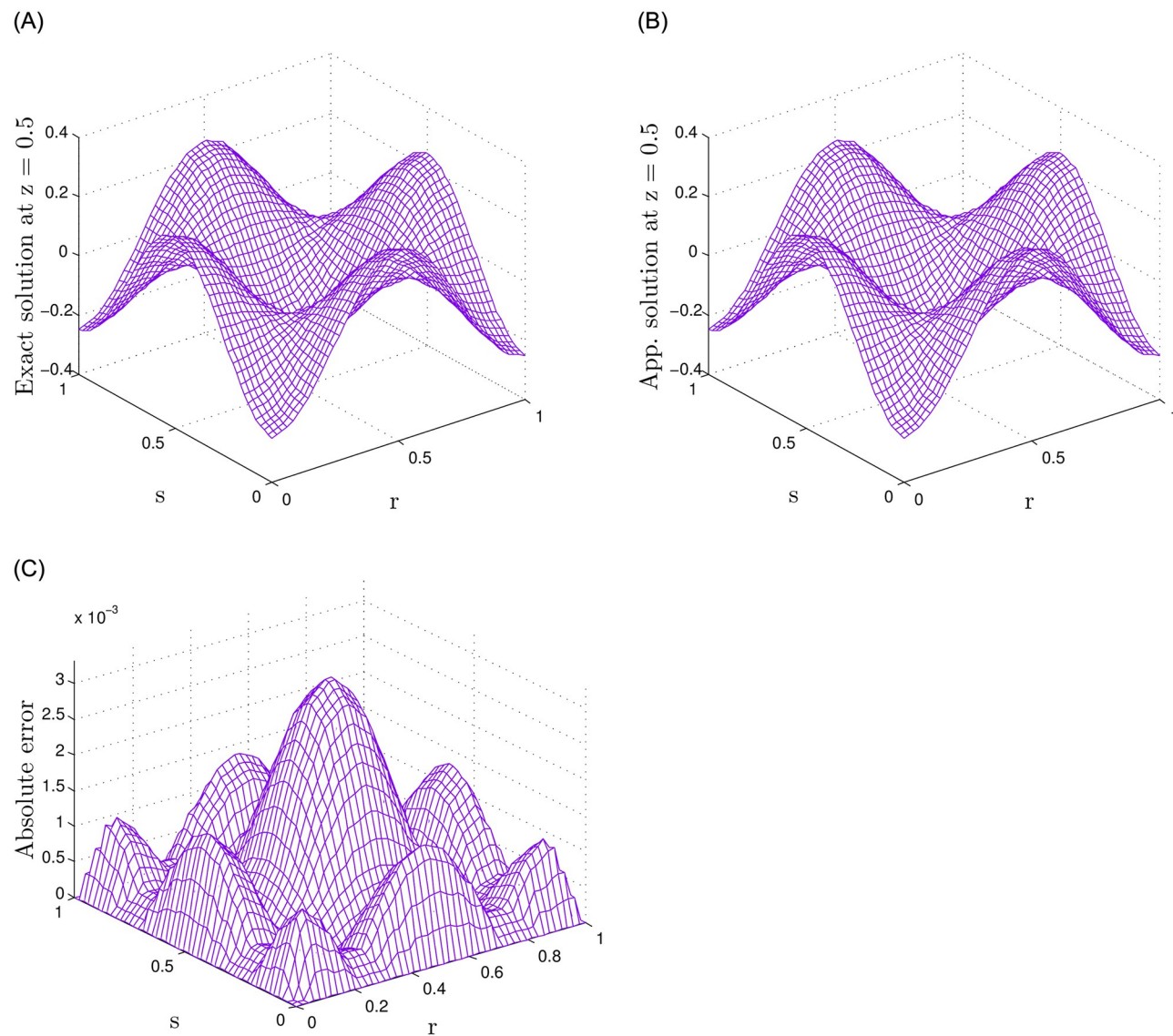

**Fig 2. For Problem 1, the results for 5-term fractional derivatives are presented, indicating a comparison between exact and numerical outcomes along with the absolute error.**

**Table 4. The outcomes obtained from utilizing the meshless approach to address Problem 2.**

| | N = 5 | | N = 10 | | N = 15 | |
|---|---|---|---|---|---|---|
| | RMS | MaxE | RMS | MaxE | RMS | MaxE |
| 2-term | 1.1440e-03 | 4.4735e-03 | 7.4621e-05 | 2.0619e-04 | 8.4585e-06 | 2.9899e-05 |
| 3-term | 1.1292e-03 | 4.4204e-03 | 7.1787e-05 | 2.0059e-04 | 8.2310e-06 | 3.1755e-05 |
| 5-term | 1.0856e-03 | 4.2558e-03 | 6.6487e-05 | 1.8873e-04 | 8.2437e-06 | 3.3623e-05 |

**Table 5. The outcomes obtained from utilizing the meshless approach to address Problem 2.**

| | | 2-term | | 3-term | | 5-term | |
|---|---|---|---|---|---|---|---|
| | $\tau$ | *RMS* | *MaxE* | *RMS* | *MaxE* | *RMS* | *MaxE* |
| $\mathfrak{J} = 0.05$ | 0.1 | 4.6709e-03 | 1.6262e-02 | 4.2843e-03 | 1.4812e-02 | 3.6234e-03 | 1.2379e-02 |
| | 0.5 | 3.9568e-03 | 1.3451e-02 | 3.9670e-03 | 1.3348e-02 | 3.8858e-03 | 1.2904e-02 |
| | 1 | 2.4894e-03 | 8.4489e-03 | 2.5477e-03 | 8.5511e-03 | 2.6069e-03 | 8.6246e-03 |
| $\mathfrak{J} = 0.005$ | 0.1 | 3.7744e-04 | 1.4124e-03 | 3.5524e-04 | 1.3144e-03 | 3.1414e-04 | 1.1418e-03 |
| | 0.5 | 3.2113e-04 | 1.1754e-03 | 3.2702e-04 | 1.1783e-03 | 3.2921e-04 | 1.1604e-03 |
| | 1 | 2.0227e-04 | 7.3829e-04 | 2.1048e-04 | 7.5517e-04 | 2.2160e-04 | 7.7584e-04 |
| $\mathfrak{J} = 0.0005$ | 0.1 | 1.3290e-05 | 3.7492e-05 | 1.2386e-05 | 3.7178e-05 | 1.0950e-05 | 3.5303e-05 |
| | 0.5 | 9.1617e-06 | 2.6956e-05 | 8.7318e-06 | 2.8228e-05 | 8.2216e-06 | 2.9913e-05 |
| | 1 | 5.5970e-06 | 1.6599e-05 | 5.3644e-06 | 1.7597e-05 | 5.1517e-06 | 1.9217e-05 |

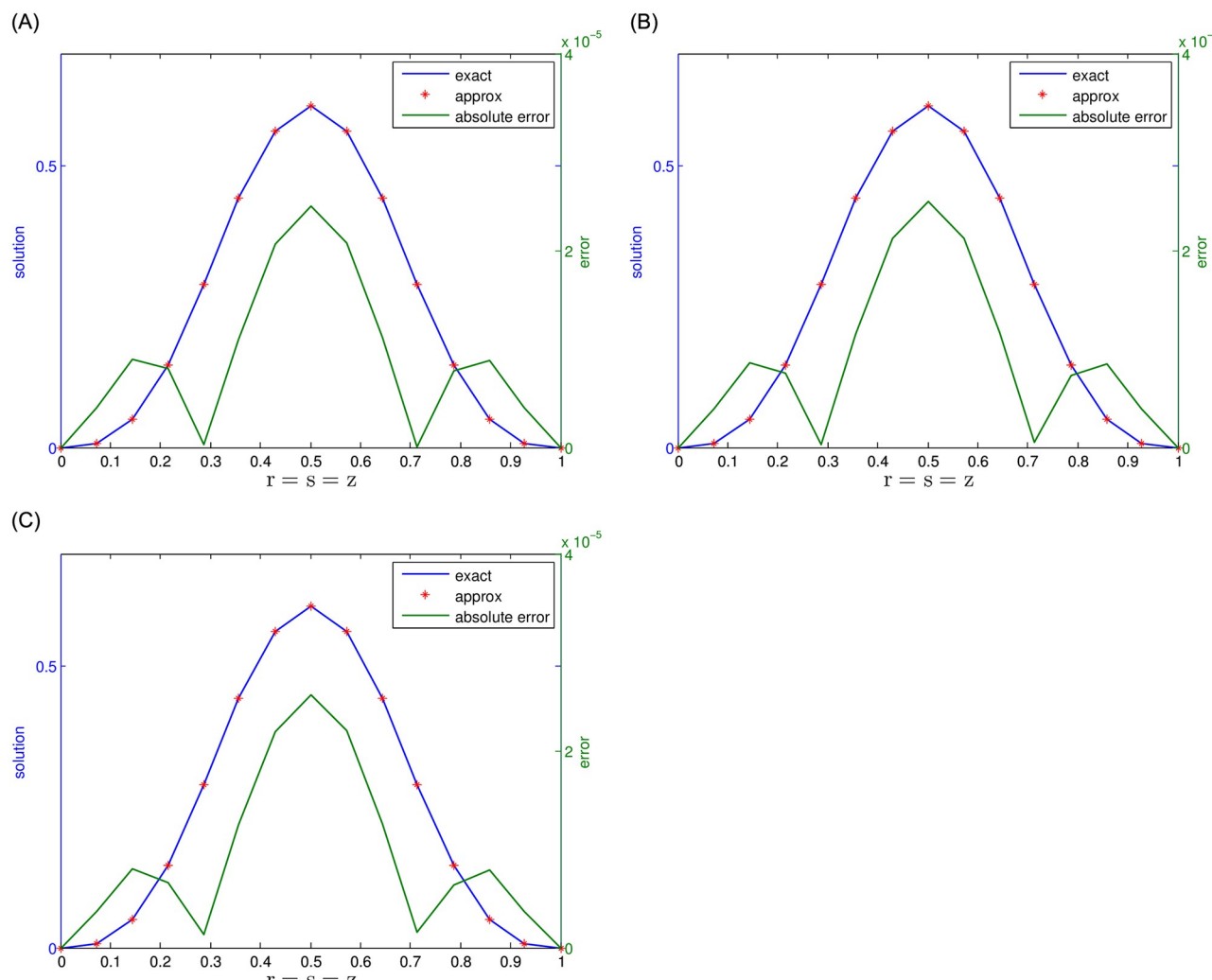

**Fig 3. For Problem 2, the results for 2-, 3-, and 5-term fractional derivatives are presented, indicating a comparison between exact and numerical outcomes along with the absolute error.**

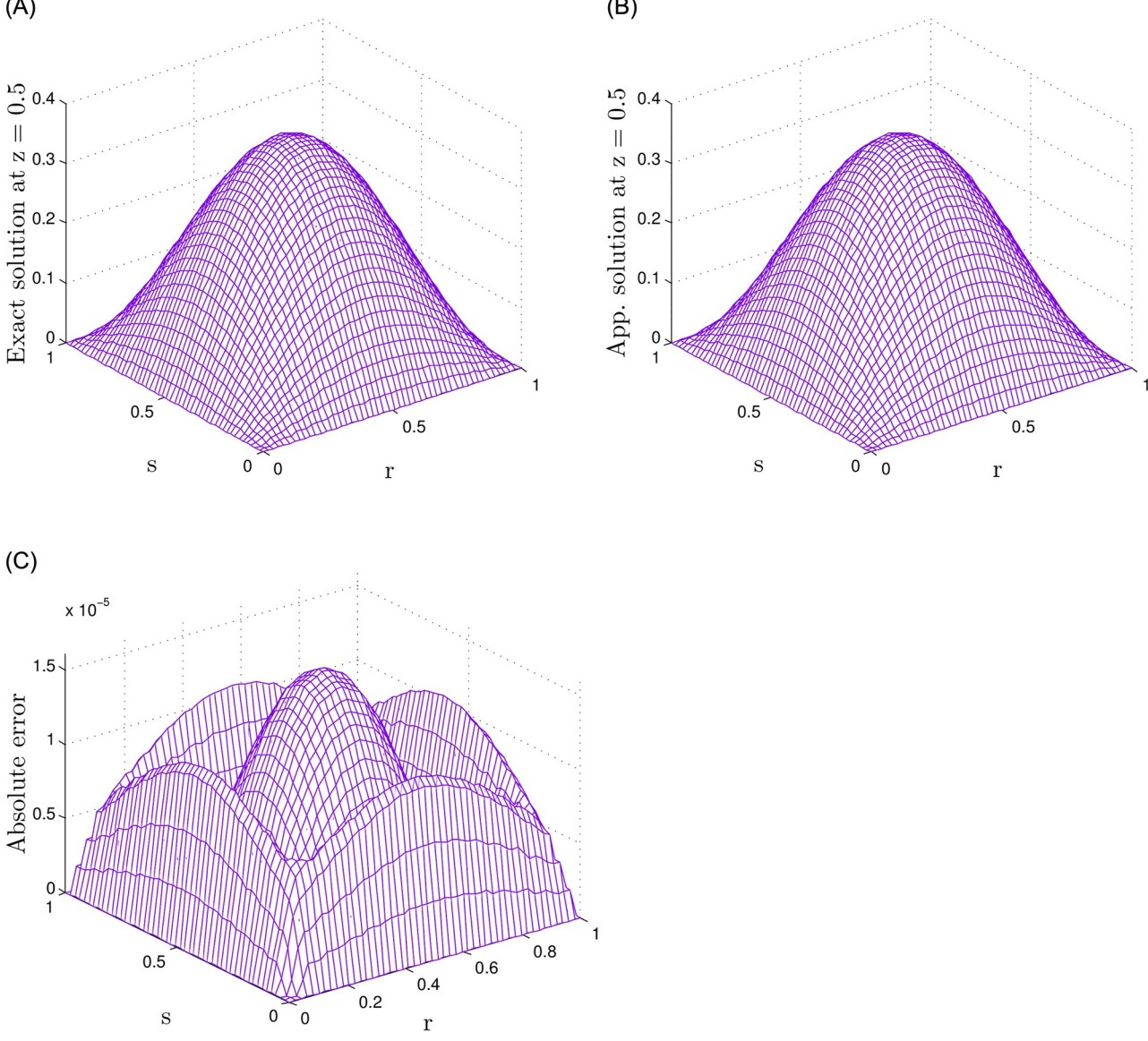

**Fig 4. For Problem 2, the results for 5-term fractional derivatives are presented, indicating a comparison between exact and numerical outcomes along with the absolute error.**

## Conclusion

The study presents a meshless computational approach for simulating the 3D multi-term time-fractional mobile-immobile diffusion equation in the Caputo sense. The methodology combines a stable Crank-Nicolson time-integration scheme with the definition of the Caputo derivative to discretize the problem in the temporal direction. The spatial function derivative is approximated using the inverse multiquadric RBF. This integration yields a sparse linear system of equations, resulting in significant reductions in computational expenses and execution time. To assess the accuracy of the proposed approach, three distinct error norms were computed. Furthermore, the effectiveness of the method was demonstrated by applying it to three different test problems with irregular computational domains. The obtained results were

(A)

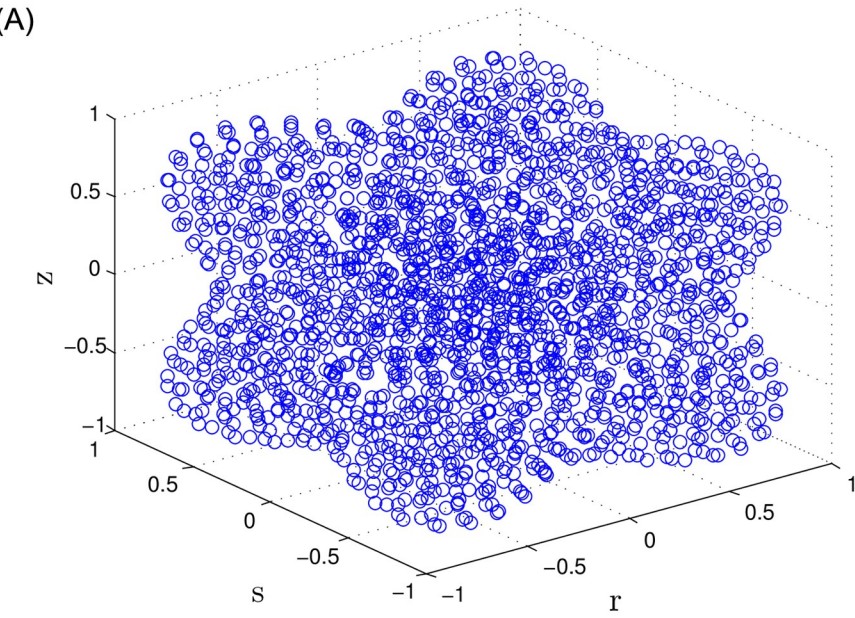

(B)

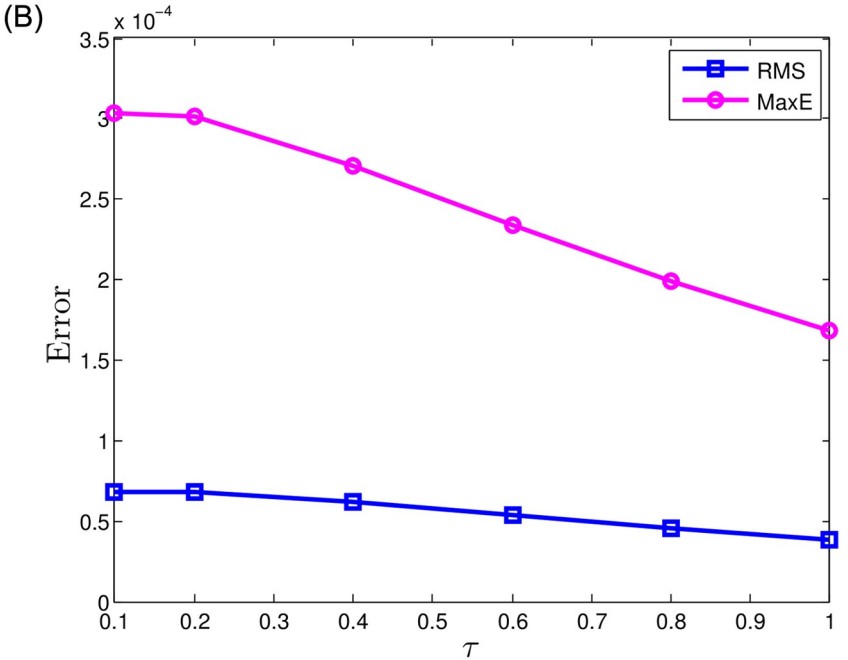

**Fig 5. For Problem 2, the results for 5-term fractional derivatives are presented for the given domain.**

subjected to rigorous evaluation, and their accuracy and efficiency were illustrated through the presentation of tables and figures. Notably, the versatility of the proposed technique enables its adaptation for various complex fractional partial differential equations with minimal modifications.

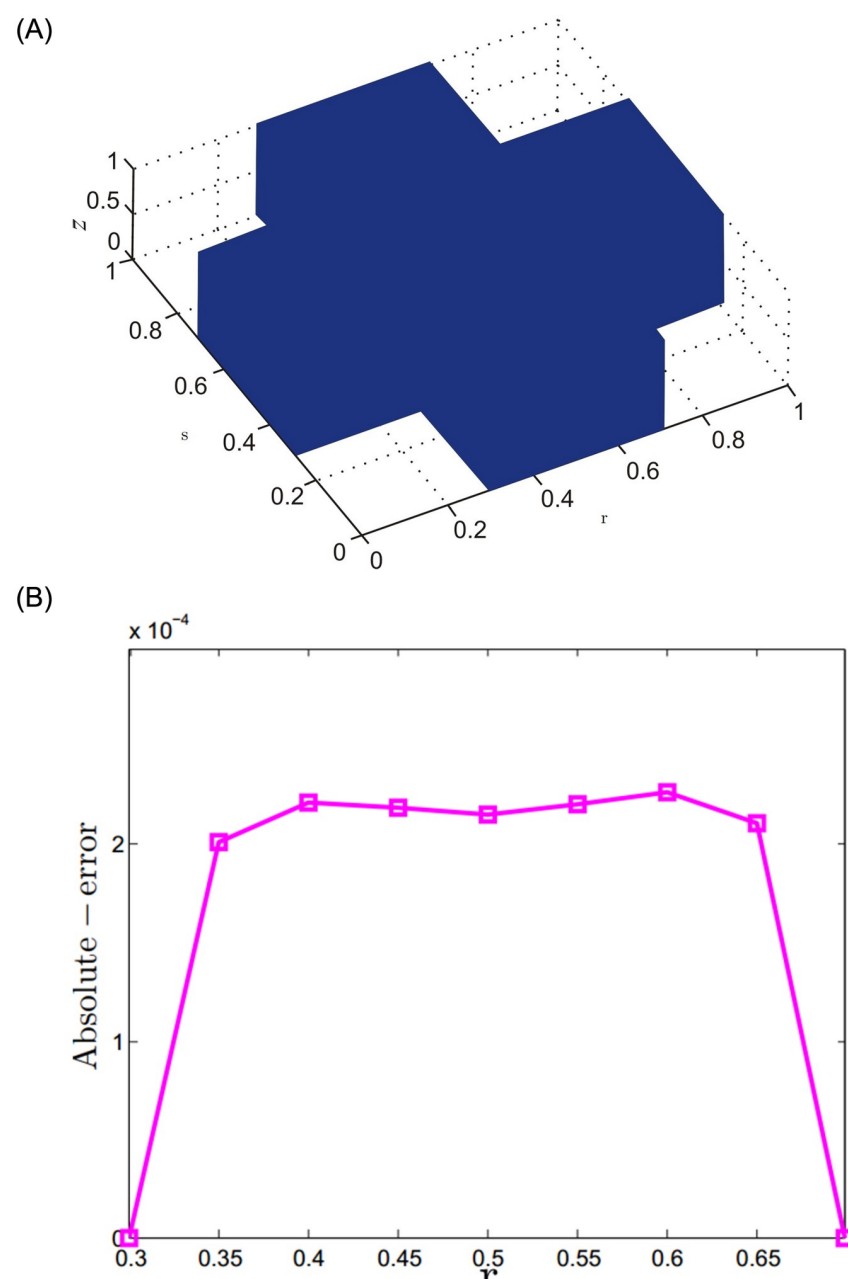

**Fig 6. For Problem 2, the results for 5-term fractional derivatives are presented for the given domain.**

**Table 6. The outcomes obtained from utilizing the meshless approach to address Problem 3.**

|  | 2-term | | 3-term | | 5-term | |
|---|---|---|---|---|---|---|
| $\gamma_k$ | RMS | MaxE | RMS | MaxE | RMS | MaxE |
| 0.15 | 8.5158e-05 | 2.2179e-04 | 8.2488e-05 | 2.1405e-04 | 7.7637e-05 | 2.0000e-04 |
| 0.35 | 8.4018e-05 | 2.1839e-04 | 8.0916e-05 | 2.0934e-04 | 7.5405e-05 | 1.9328e-04 |
| 0.65 | 8.6282e-05 | 2.2303e-04 | 8.4421e-05 | 2.1658e-04 | 8.1384e-05 | 2.0867e-04 |
| 0.85 | 1.0965e-04 | 2.8202e-04 | 1.1841e-04 | 3.0489e-04 | 1.3439e-04 | 3.4600e-04 |

**Table 7. The outcomes obtained from utilizing the meshless approach to address Problem 3.**

| | | 2-term | | 3-term | | 5-term | |
|---|---|---|---|---|---|---|---|
| | N | RMS | MaxE | RMS | MaxE | RMS | MaxE |
| $\mathfrak{J} = 0.05$ | 5 | 3.7026e-03 | 2.6843e-02 | 2.4127e-03 | 1.8632e-02 | 1.4783e-03 | 8.2977e-03 |
| | 8 | 4.8860e-03 | 3.9300e-02 | 3.2919e-03 | 2.9620e-02 | 1.8620e-03 | 1.7349e-02 |
| | 12 | 4.7913e-03 | 4.2270e-02 | 3.2120e-03 | 3.2692e-02 | 1.8705e-03 | 2.0483e-02 |
| $\mathfrak{J} = 0.005$ | 5 | 6.7672e-04 | 2.2190e-03 | 6.6380e-04 | 2.1495e-03 | 6.3982e-04 | 2.0231e-03 |
| | 8 | 2.3204e-04 | 1.5766e-03 | 2.2649e-04 | 1.3703e-03 | 2.2816e-04 | 1.0408e-03 |
| | 12 | 2.3145e-04 | 2.0574e-03 | 1.9906e-04 | 1.8409e-03 | 1.5776e-04 | 1.4947e-03 |
| $\mathfrak{J} = 0.0005$ | 5 | 8.8448e-04 | 2.6489e-03 | 8.3706e-04 | 2.4804e-03 | 7.5605e-04 | 2.1959e-03 |
| | 8 | 3.5360e-04 | 9.9279e-04 | 3.3594e-04 | 9.3576e-04 | 3.0569e-04 | 8.3844e-04 |
| | 12 | 1.4388e-04 | 3.7946e-04 | 1.3755e-04 | 3.6262e-04 | 1.2665e-04 | 3.3338e-04 |

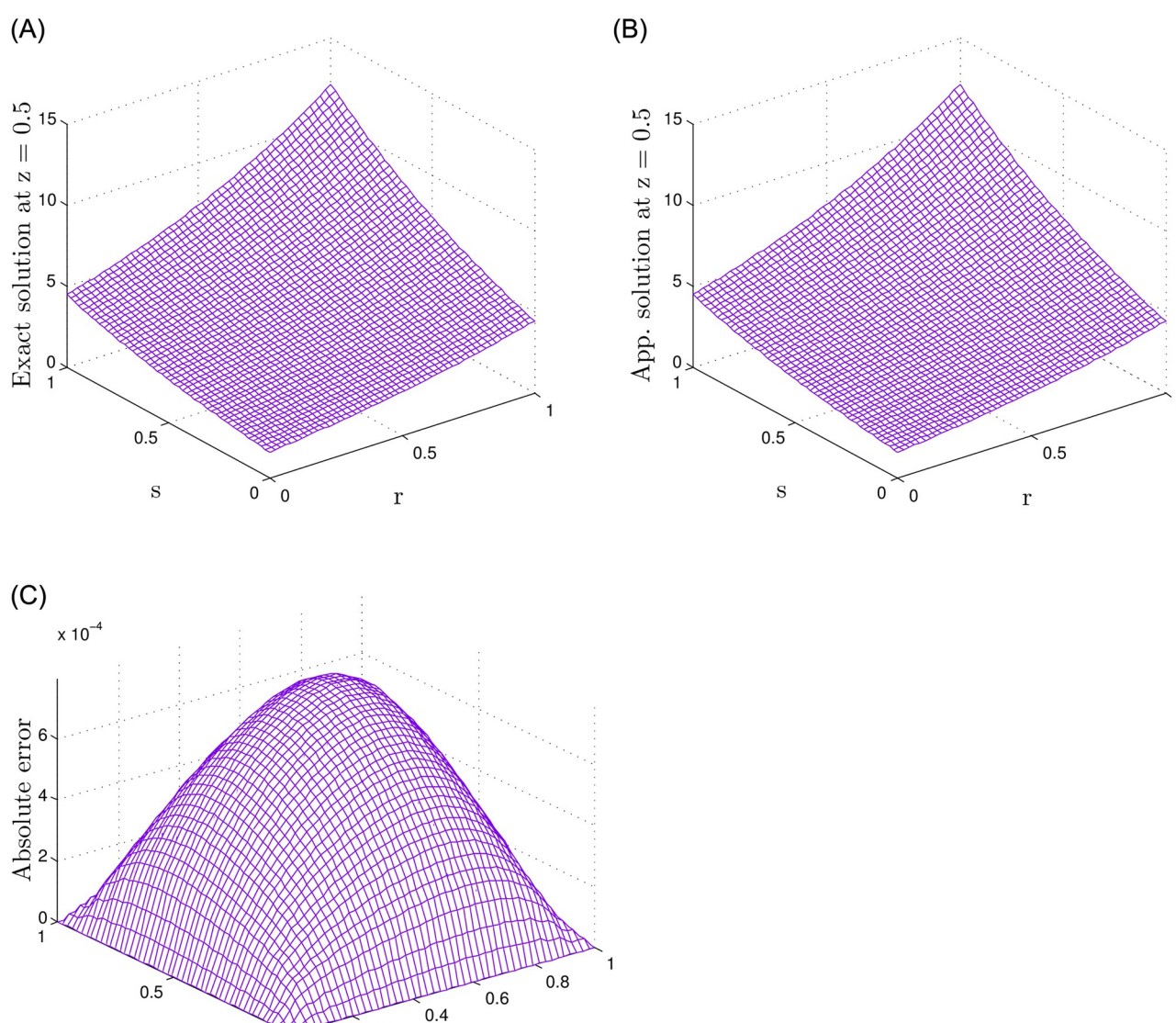

**Fig 7. For Test Problem 3, the results for 5-term fractional derivatives are presented, indicating a comparison between exact and numerical outcomes along with the absolute error.**

## Author Contributions

**Conceptualization:** Imtiaz Ahmad.

**Data curation:** Rashid Jan.

**Formal analysis:** Rashid Jan.

**Funding acquisition:** Mohamed Mousa.

**Methodology:** Imtiaz Ahmad, Ihteram Ali.

**Software:** Imtiaz Ahmad, Rashid Jan.

**Validation:** Rashid Jan.

**Writing – original draft:** Imtiaz Ahmad.

**Writing – review & editing:** Ihteram Ali, Sahar Ahmed Idris, Mohamed Mousa.

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
