## [Decision Letter · Decision Letter 0]

18 Sep 2023

PONE-D-23-26366Solutions of a three-dimensional multi-term fractional anomalous solute transport model for contamination in groundwaterPLOS ONE

Dear Dr. Ahmad,

Thank you for submitting your manuscript to PLOS ONE. After careful consideration, we feel that it has merit but does not fully meet PLOS ONE’s publication criteria as it currently stands. Therefore, we invite you to submit a revised version of the manuscript that addresses the points raised during the review process.

We look forward to receiving your revised manuscript.

Kind regards,

Muhammad Nadeem

Academic Editor

PLOS ONE

ournal requirements:

[Research Supporting Project number (RSP2023R167), King Saud University, Riyadh, Saudi Arabia.]

 [The author(s) received no specific funding for this work.]

4. We note that Figure 2, 4, 5 and 7 in your submission contain copyrighted images. All PLOS content is published under the Creative Commons Attribution License (CC BY 4.0), which means that the manuscript, images, and Supporting Information files will be freely available online, and any third party is permitted to access, download, copy, distribute, and use these materials in any way, even commercially, with proper attribution. For more information, see our copyright guidelines: http://journals.plos.org/plosone/s/licenses-and-copyright.

A. You may seek permission from the original copyright holder of Figure 2, 4, 5 and 7 to publish the content specifically under the CC BY 4.0 license. 

B. If you are unable to obtain permission from the original copyright holder to publish these figures under the CC BY 4.0 license or if the copyright holder’s requirements are incompatible with the CC BY 4.0 license, please either i) remove the figure or ii) supply a replacement figure that complies with the CC BY 4.0 license. Please check copyright information on all replacement figures and update the figure caption with source information. If applicable, please specify in the figure caption text when a figure is similar but not identical to the original image and is therefore for illustrative purposes only.

Additional Editor Comments:

According to the comments of the reviewer, your paper needs major revisions. Please, submit your revised version at your earliest convenience.

Reviewers' comments:

Reviewer's Responses to Questions

**Comments to the Author**

1. Is the manuscript technically sound, and do the data support the conclusions?

Reviewer #1: Yes

Reviewer #2: Yes

Reviewer #3: Yes

2. Has the statistical analysis been performed appropriately and rigorously? 

Reviewer #1: N/A

Reviewer #2: Yes

Reviewer #3: N/A

3. Have the authors made all data underlying the findings in their manuscript fully available?

Reviewer #1: Yes

Reviewer #2: Yes

Reviewer #3: Yes

4. Is the manuscript presented in an intelligible fashion and written in standard English?

Reviewer #1: Yes

Reviewer #2: Yes

Reviewer #3: Yes

5. Review Comments to the Author

Reviewer #1: The paper considers an interesting problem and its solution is presented adequately. The results presented are new and appealing. The paper can be accepted for publication provided the comments are incorpurated.

Reviewer #2: The manuscript is well-written, and the results appear to be accurate and genuine. Prior to considering publication in this esteemed journal, the following comments should be addressed, please see the attachment.

Reviewer #3: REPORT ON: Solutions of a three-dimensional multi-term fractional anomalous solute transport model for contamination in groundwater

This paper deals with the numerical solution of three-dimensional multi-term time-fractional mobile-immobile diffusion equation in the Caputo sense using meshless computational approach. The methodology combines a stable Crank-Nicolson time-integration scheme with the definition of the Caputo derivative to discretize the problem in the temporal direction. The results of the paper are considerable. But the manuscript must be revised in the view of following comments.

There are many the misprints/typos. I suggest authors to read the paper carefully to avoid such typos. Further, use proper punctuation.

The novelty and significance of the results are not clear in view of the other existing literature. The authors should emphasize which aspects of this paper are novel compared to the other existing results in the available literature.

The $h$ and the N in the definition of the RMS norm are ambiguous, until now they have been both used as indexes. Please relate these norms with the notation used in previous in a consistent way. What is the space-step size $h$ in a method without mesh?

Introduction should be updated with the latest work on FDEs.

Revised the caption of Figures 1. Furthermore, check all the captions as well.

Please check all the punctuations at the end of mathematical work.

6. PLOS authors have the option to publish the peer review history of their article (what does this mean?). If published, this will include your full peer review and any attached files.

Reviewer #1: **Yes: **Dr. Mehnaz Shakeel

Reviewer #2: No

Reviewer #3: No

---

## [Author Response · Author response to Decision Letter 0]

21 Oct 2023

Author's Response to Reviewer/Editor Comment

Dear Editor/Reviewer, 

We are thankful to honorable Editor and the worthy reviewers for their generous time to review and giving valuable comments as well as useful suggestions on our manuscript that help us to improve the quality of the paper and make the paper clearer to the readers. We have tried our best to answer all the queries/suggestions from the reviewers, hope our efforts will meet the criteria of the reviewers. All the suggestions have been incorporated in the revised version of manuscript at the appropriate places and highlighted in blue color. We have modified the manuscript accordingly, and the detailed corrections are listed below point by point.

Reviewer # 1: Comments and Response 

The paper considers an interesting problem and its solution is presented adequately. The results presented are new and appealing. The paper can be accepted for publication provided the following comments:

Comment 1: Even though the topic is really interesting and the paper have the potential to be a nice add to the literature, I found the manuscript really difficult to read. Basically, the paper needs text editing and a re-organization of the data presented.

Response 1: We wish to acknowledge the anonymous referees whose suggestions have helped improve the quality of the paper. We have corrected some typographical mistakes and a language expert is consulted in the revised version. The paper is re-organized and the title of the paper has been modified.

Comment 2: What is the shape parameter “c"? What are the effects of this free parameter on the solution accuracy? 

Response 2: Most Radial Basis Functions depend on a shape parameter ‘c’, with ‘c’ approach to 0 corresponding to the limit of increasing flatness. Lowering the value of ‘c’ usually increases the resulting accuracy to some point. Due to scaling the local meshless methods is not so sensitive to the selection of the shape parameter as compare to global meshless methods. Some optimization techniques are also in practice to find the better value of c. Finding an optimal value of shape parameter c is still unknown and is currently under rigorous investigation. It is discovered that the value of shape parameter c also depends upon the problem.

Comment 3: List some properties of the Meshless method.

Response 3: The suggestion has been incorporated and salient features of the Meshless method have been included.

Comment 4: Describe the following in details: Radial basis function, Mobile-immobile, and solute transport model.

Response 4: We have incorporated the suggestions and include details of method, RBFs and about the underlying model. 

Comment 5: Why the presented the technique approximate the solution on set of uniform and scattered nodes and it leads to a sparse and well-conditioned coefficient matrix.

Response 5: This is for the information of honorable referee that the beauty of local meshless method is to approximate the solution on set of uniform and scattered nodes and it leads to a sparse well-conditioned coefficient matrix. The full detail is available in the second last paragraph of the paper in introduction section.

Comment 6: Abbreviations should be defined when use at first time in the paper.

Response 6: The suggestions have been incorporated.

Comment 7: The graphical visualization need to be express in clearer way.

I am trying to understand the behaviour of the graphs but the given information according to my point of view is not sufficient share the source how we visualize the graphs and check their accuracy. So, paste all computational work here.

Response 7: The suggestion of the honorable reviewer has been incorporated. We have provided more details regarding all numerical results specially the graphical representations. 

Comment 8: The punctuations at the end of the equations are missing.

Response 7: The suggestions have been incorporated.

Reviewer # 2: Comments and Response 

The paper is well written and the results looks corrects and genuine. Before accepting this article in this esteem journal the following comments must be incorporated:

Comment 1: While the topic is intriguing and the paper holds the potential to contribute significantly to the literature, certain improvements are needed. It is imperative that the abstract clearly elucidates the problem studied, its significance, and the employed methodology. The paper contains several typos and errors, and foremost, a thorough revision of the English language is essential.

Response 1: We wish to acknowledge the anonymous referees whose suggestions have helped improve the quality of the paper. We have corrected some typographical mistakes and a language expert is consulted in the revised version. We have incorporated the suggestions. The abstract, introduction and the conclusions have been modified accordingly.

Comment 2: The Introduction should compellingly justify the study's utility and delineate its novelty or originality. This should involve discussing existing knowledge in the open literature and identifying gaps.

Response 2: The suggestions have been incorporated.

Comment 3: The rationale behind employing the presented technique for approximating solutions on both uniform and scattered nodes, leading to a sparse and well-conditioned coefficient matrix, should be clarified.

Response 3: This is for the information of honorable referee that the beauty of local meshless method is to approximate the solution on set of uniform and scattered nodes and it leads to a sparse well-conditioned coefficient matrix. The full detail is available in introduction section.

Comment 4: Further elaboration is required to explain why the technique approximates solutions on sets of uniform and scattered nodes, ultimately yielding a sparse and well-conditioned coefficient matrix.

Response 4: This is for the information of honorable referee that the beauty of local meshless method is to approximate the solution on set of uniform and scattered nodes and it leads to a sparse well-conditioned coefficient matrix. The full detail is available in the second last paragraph of the paper in introduction section.

Comment 5: The authors are encouraged to provide additional details regarding their original contributions throughout the manuscript.

Response 5: The suggestion has been implemented and the contribution have been added in introduction.

Comment 6: It is recommended to ensure proper punctuation (comma or period) is consistently used at the end of equations, adhering to typing conventions. A comprehensive review of the entire paper for such errors is necessary.

Response 6: We check the whole manuscript for punctuations and English language. 

Comment 7: Enhanced descriptions of the examples would enhance clarity.

Comment 8: Clear definition of symbols and proper equation numbering are essential.

Response 7-8: The suggestions have been incorporated. 

Reviewer # 3: Comments and Response 

This paper deals with the numerical solution of three-dimensional multi-term time-fractional mobile-immobile diffusion equation in the Caputo sense using meshless computational approach. The methodology combines a stable Crank-Nicolson time-integration scheme with the definition of the Caputo derivative to discretize the problem in the temporal direction. The results of the paper are considerable. But the manuscript must be revised in the view of following comments.

Comment 1: Please note that these are not only the misprints/typos. I suggest authors to read the paper carefully to avoid such typos. Further, use proper punctuation.

Response 1: We have read the whole manuscript carefully and corrected the typos and punctuation mistakes.

Comment 2: The novelty and significance of the results are not clear in view of the other existing literature. The authors should emphasize which aspects of this paper are novel compared to the other existing results in the available literature. 

Response 2: We hope that the honorable referee will be satisfied from the revised version. The novelty of the paper is the coupling of Caputo time fractional technique with local meshless method for the solution of multi-term time-fractional PDE models. Another contribution of the paper is that the local meshless methods have been shown more stable and efficient in the context multi-term time-fractional PDE models than their global counter parts. 

In this article, the Crank-Nicolson time discretization schemes are coupled with the local meshless method for the numerical solution of the proposed model. Inverse multiquadric radial basis functions are considered. Furthermore, one irregular domains are also considered in numerical examinations. 

The accuracy and stability of the meshless based on RBFs fully depend on the value of shape parameter c as well as the numbers of nodes N. It is observed from literature that the accuracy and conditional number of the global meshless method are extremely sensitive to the values of c. The paper will be useful for a lot of scientists working in new trends of meshless methods in term of multi-term time-fractional PDEs. The present work will lay a foundation for numerical solution of such types of PDEs that have been got broadly attention in almost all disciplines of engineering and, then, possesses biological implications and extends the former researches.

Unlike Traditional numerical methods, such as finite element, finite difference, or finite volume methods, meshless methods use radial distance r to realize numerical solution of the problem. This is achieved by composing some univariate basic function with a (Euclidean) norm, and therefore turning a problem involving many space dimensions into one that is virtually one-dimensional.

So in the case of meshless methods going from one –dimensional case to higher dimensional (2-, 3-, 4- dimensions etc.) is not a big deal. The method structure remains the same. Only the size of the matrix increases. The only change which is needed is the change in the radial function.

In one dimensional case, the radial function is defined as r(x)=√((x-x_i )^2+c^2 ) . In three- dimensional case it becomes r(x,y,z)=√((x-x_i )^2+(y-y_j )^2+(z-z_k )^2+c^2 ) and so on. Hence, irrespective of the dimensionality of the problem, we are dealing with one variable r. Of course, the size of the matrix increases in higher dimensions but the structure the method is preserved. This is one of the advantage of meshless method over the other traditional numerical methods.

Comments 3: The $h$ and the N in the definition of the RMS norm are ambiguous, until now they have been both used as indexes. Please relate these norms with the notation used in previous in a consistent way. What is the space-step size $h$ in a method without mesh?

Response 3: The suggestion has been incorporated and we have corrected the definitions and relate the notation according to previous section.

The meshfree methods do not require connection between nodes of the simulation domain that is MESH, but are rather based on interaction of each node with all its neighbors. The space-step size is used in all meshless methods to define nodal points in the computational domain. In contrast, in the mesh-based methods, each point has a fixed number of predefined neighbors, and this connectivity between neighbors can be used to define mathematical operators.

Comment 4: Introduction can be updated with the following latest work on FDEs.

Response 4: We have included the appropriate references and some other latest references have been included as well.

Comment 5: Revised the caption of Figures 1. Furthermore, check all the captions as well.

Response 5: The suggestions has been incorporated.

Comment 6: Please check all the punctuations at the end of mathematical work.

Response 6: We have check the whole manuscript for punctuations and other typos mistakes.

Comment 7: Arrange the references in the same format (adhering to the journal style).

Response 7: We have incorporated the suggestion.

Thank you so much for your valuable suggestions. The required changes have been made in the revised manuscript. Finally, we thank again the reviewers for their comments that led to improve our paper. We are looking forward to your positive response. 

---

## [Decision Letter · Decision Letter 1]

31 Oct 2023

Solutions of a three-dimensional multi-term fractional anomalous solute transport model for contamination in groundwater

PONE-D-23-26366R1

Dear Dr. Ahmad,

We’re pleased to inform you that your manuscript has been judged scientifically suitable for publication and will be formally accepted for publication once it meets all outstanding technical requirements.

Kind regards,

Muhammad Nadeem

Academic Editor

PLOS ONE

Additional Editor Comments (optional):

Reviewers' comments:

Reviewer's Responses to Questions

**Comments to the Author**

1. If the authors have adequately addressed your comments raised in a previous round of review and you feel that this manuscript is now acceptable for publication, you may indicate that here to bypass the “Comments to the Author” section, enter your conflict of interest statement in the “Confidential to Editor” section, and submit your "Accept" recommendation.

Reviewer #2: All comments have been addressed

Reviewer #3: (No Response)

2. Is the manuscript technically sound, and do the data support the conclusions?

Reviewer #2: Yes

Reviewer #3: (No Response)

3. Has the statistical analysis been performed appropriately and rigorously? 

Reviewer #2: Yes

Reviewer #3: (No Response)

4. Have the authors made all data underlying the findings in their manuscript fully available?

Reviewer #2: Yes

Reviewer #3: (No Response)

5. Is the manuscript presented in an intelligible fashion and written in standard English?

Reviewer #2: Yes

Reviewer #3: (No Response)

6. Review Comments to the Author

Reviewer #2: All my comments are well consdiered and the whole manuscript is good. The manuscript can be accepted in the present form.

Reviewer #3: (No Response)

7. PLOS authors have the option to publish the peer review history of their article (what does this mean?). If published, this will include your full peer review and any attached files.

Reviewer #2: No

Reviewer #3: No

---

## [Editor Report · Acceptance letter]

29 Nov 2023

PONE-D-23-26366R1 

Solutions of a three-dimensional multi-term fractional anomalous solute transport model for contamination in groundwater 

Dear Dr. Ahmad:

I'm pleased to inform you that your manuscript has been deemed suitable for publication in PLOS ONE. Congratulations! Your manuscript is now with our production department. 

Kind regards, 

on behalf of

Dr. Muhammad Nadeem 

Academic Editor

PLOS ONE